# Uncovering the genomic and metagenomic research potential in old ethanol-preserved snakes

Claus M. Zacho[1]☯, Martina A. Bager[2]☯, Ashot Margaryan[2,3], Peter Gravlund[4], Anders Galatius[5], Arne R. Rasmussen[6], Morten E. Allentoft[1,7]*

1 Lundbeck Foundation GeoGenetics Centre, GLOBE Institute, University of Copenhagen, Copenhagen, Denmark, 2 Section for EvoGenomics, GLOBE Institute, University of Copenhagen, Copenhagen, Denmark, 3 Center for Evolutionary Hologenomics, University of Copenhagen, Copenhagen, Denmark, 4 Museum of Eastern Zealand, Faxe, Denmark, 5 Department of Bioscience, Aarhus University, Roskilde, Denmark, 6 Institute of Conservation, Royal Danish Academy—Architecture, Design, Conservation, Copenhagen, Denmark, 7 Trace and Environmental DNA (TrEnD) Laboratory, School of Molecular and Life Sciences, Curtin University, Perth, Australia

☯ These authors contributed equally to this work.
* morten.allentoft@curtin.edu.au

**Data Availability Statement:** Sequence data will be available at the European Nucleotide Archive (ENA) under accession number PRJEB46456 the 19 of August 2021.

## Abstract

Natural history museum collections worldwide represent a tremendous resource of information on past and present biodiversity. Fish, reptiles, amphibians and many invertebrate collections have often been preserved in ethanol for decades or centuries and our knowledge on the genomic and metagenomic research potential of such material is limited. Here, we use ancient DNA protocols, combined with shotgun sequencing to test the molecular preservation in liver, skin and bone tissue from five old (1842 to 1964) museum specimens of the common garter snake (*Thamnophis sirtalis*). When mapping reads to a *T. sirtalis* reference genome, we find that the DNA molecules are highly damaged with short average sequence lengths (38–64 bp) and high C-T deamination, ranging from 9% to 21% at the first position. Despite this, the samples displayed relatively high endogenous DNA content, ranging from 26% to 56%, revealing that genome-scale analyses are indeed possible from all specimens and tissues included here. Of the three tested types of tissue, bone shows marginally but significantly higher DNA quality in these metrics. Though at least one of the snakes had been exposed to formalin, neither the concentration nor the quality of the obtained DNA was affected. Lastly, we demonstrate that these specimens display a diverse and tissue-specific microbial genetic profile, thus offering authentic metagenomic data despite being submerged in ethanol for many years. Our results emphasize that historical museum collections continue to offer an invaluable source of information in the era of genomics.

## Introduction

Natural history museum collections represent a tremendous resource for studying past and present biodiversity on our planet [1]. For centuries, researchers and naturalists have deposited

**Funding:** MEA is funded by the Independent Research Fund Denmark (Sapere Aude, grant 7027-00147B) - https://dff.dk/en/front-page?set_language=en. The funders had no role in study design, data collection, and analysis, decision to publish, or preparation of the manuscript.

**Competing interests:** The authors have declared that no competing interests exist.

specimens in such collections worldwide [e.g., 2, 3], often with the aim of studying and describing variation between and within species based on detailed morphological investigations. However, the many millions of organisms preserved with precise information on collection date and locality can be equally relevant for studying species phylogenetics [4, 5], temporal patterns (for example in relation to changes in species distributions or composition) [6–8], trends in pollution-levels [9–11], or for documenting changes to the gene pools of endangered, extinct, or locally-extinct species [e.g., 12–17]. Examining DNA sequences from a species or population sampled at different points in space and time offer a plethora of analytical possibilities, allowing us to understand microevolutionary responses to our rapidly changing biosphere. Obtaining such genetic measures of the past can be important in conservation contexts and aid in present day management of threatened species [18, 19]. Moreover, museum collections often represent much easier access to tissue samples compared to getting new samples from individuals in the field. This is particularly important for population-scale projects where many individuals must be sampled to cover the genetic diversity in a given gene pool. Access to a museum collection allows the researcher to reduce fieldwork costs and effort, where the outcome is often unpredictable.

However, in a molecular context museum material has one major disadvantage compared to freshly obtained tissue: poor DNA preservation. DNA is an unstable molecule which degrades *post-mortem*, leaving the DNA strands progressively more fragmented over time [e.g., 20, 21]. Furthermore, the state of molecular preservation in a given specimen is likely to be heavily affected by the mode of preservation. Samples obtained from museum collections rarely involve fresh tissue sampled specifically for genetic analysis. Instead, samples are often stored in ethanol as is the case for a wide range of organisms including mammals, fish, amphibians, reptiles and various invertebrate taxa. Ethanol preservation is widely used in museum collections, as this is an affordable and non-toxic conservation agent that preserves samples against microbial activity and autolysis, thus ultimately enhancing the specimen's longevity [22]. Because it reduces microbial and hydrolytic activity, ethanol preservation will also slow down the degradation of the DNA molecules, but it cannot completely prevent a slow and steady decay, in particular if the ethanol purity is low (i.e. below 70%). Therefore, it is expected that old museum specimens in ethanol will display degraded DNA molecules.

Formalin fixation adds another highly problematic component to collection-based DNA research. Formalin has been used as a fixation agent on biological samples since the early 1890s [23, 24] and has routinely been used in museum collections. Here the agent is usually injected into the specimen and/or it is being soaked in a 10% (or higher) formalin solution for up to several weeks. The treatment hardens the tissue by generating cross-links between amino acids in protein molecules [23–25]. Thus, the fixation maintains the body shape of the specimen, which is crucial for morphological studies [26]. However, the treatment also creates cross-links between amino acids and DNA and RNA [27, 28] which will complicate later DNA extraction and introduce polymerase "blocks" that prevent successful PCR reactions [27]. Additionally, an unbuffered formalin treatment can lead to very low pH (<1), resulting in rapid fragmentation of the DNA [27]. Inevitably, the effects of formalin exposure, combined with prolonged ethanol storage, will result in degraded molecules that complicate genetic analyses. To overcome these challenges, a number of different protocol improvements have been proposed [e.g., 14, 27–29]. Moreover, the switching to Next Generation Sequencing (NGS) has enabled the sequencing of short degraded molecules in museum material, where PCR-based technologies often fail [12, 13, 19, 30–40].

In ancient DNA (aDNA) research, challenges with short and degraded molecules can reach extreme levels. Therefore, when working with problematic museum specimens, it may be fruitful to apply methodological advances that have occurred in the aDNA field in recent years. In

our current era of genomics and NGS, the importance of being very selective when sampling ancient specimens has become increasingly clear. By obtaining samples with a high endogenous DNA content (i.e., the proportion of DNA molecules deriving from the target species), the amount of sequencing required to reach a desired level of genomic coverage can be reduced by orders of magnitude. However, the endogenous DNA content for many ancient samples has proven to be very low (often <1%), and represents only a minor fraction of the total quantity of sequences, in which the majority are microbial DNA [e.g., 41]. Thus, many recent aDNA efforts have been aimed at maximizing the endogenous DNA content through improved sampling [e.g., 42, 43], improved DNA extraction [44–46], and improved library preparation protocols [47, 48]. Importantly, extensive differences have been found when comparing endogenous DNA content in different types of bones [42, 43, 49] or even within different compartments of the same tooth [45], highlighting the importance of very careful substrate selection.

By recognizing the obvious similarities between museum-based DNA and aDNA research, it seems likely that similar insights could be gained by testing DNA preservation in different tissue types in old museum specimens. In the current study, we employed an ancient DNA framework to test the DNA quality in different tissue types sampled from snakes that have been preserved in ethanol for many years. Because the shotgun sequencing approach we use here is not restricted to DNA of the target species, it can also provide a genetic profile of the microbial diversity in the tissue sample. Such metagenomic analyses have gained considerable attention in the aDNA community in recent years, uncovering information on past microbial communities and prehistoric pathogen infections in humans [50–58]. To our knowledge, very few studies have investigated if the degraded DNA in old ethanol-preserved museum specimens represents a similar 'goldmine' of metagenomic information [albeit see 59, 60]. In this study we will therefore explore this potential by examining taxonomic diversity among the millions of sequences that cannot be identified as snake DNA.

In summary, this investigation employs an aDNA methodology to systematically explore the genomic and metagenomic potential in different types of tissue (liver, skin, bone) from old ethanol-stored museum specimens. We used a silica-in-solution extraction method, which has proven highly effective in recovering very short DNA fragments [44], combined with NGS shotgun sequencing. The many millions of generated sequences were bioinformatically mapped against a suitable reference genome, allowing us to statistically explore the state of molecular preservation. We also applied an enzymatic test [61] to assess if any of these specimens had previously been exposed to formalin as part of their preservation treatment.

When performing destructive sampling of valuable museum material, it is imperative to balance the pros and cons and use an approach that is optimized towards the specific research aims. As such, our presented results add to a growing number of observations that will ultimately allow us to establish best practice in terms of sampling, chemical processing, and sequencing of biological museum specimens.

## Methods

### Samples

Five *Thamnophis sirtalis* (common garter snake) specimens stored in the collections of the Natural History Museum of Denmark were used for this study (Table 1). This species was selected because of the availability of a reference genome (see below), allowing for detailed bioinformatic assessment of the DNA quality. All specimens were collected between 1842 and 1964. Two of the specimens were originally collected in the US, reflecting the natural distribution of *T. sirtalis*, which ranges from Mexico, throughout the United States and southern

**Table 1. Five specimens of *Thamnophis sirtalis* included in this study.**

| Museum # | Collection year | Location | Country |
|---|---|---|---|
| ZMUC R602799 | 1842 | Pennsylvania, West | USA |
| ZMUC R602815 | 1880 | Shoalwater Bay | USA |
| ZMUC R602826 | 1920 | Zoo, Copenhagen | Denmark |
| ZMUC R602827 | 1923 | Zoo, Copenhagen | Denmark |
| ZMUC R603729 | 1964 | Bornholm | Denmark |

The Danish island Bornholm is not part of the distribution range for this species, so similar to the zoo animals from Copenhagen, we assume that this must be an individual that died in captivity.

Canada [62]. Three specimens were collected in Denmark; two were from Copenhagen Zoo, whereas the last individual was collected on the island of Bornholm. Since *T. sirtalis* is not native to Denmark, it is assumed that the specimen from Bornholm died in captivity. Three samples were obtained from each specimen: a piece of liver (c. 100 mg), a piece of skin with muscle attached (c. 100 mg), and a piece of the vertebral column (c. 50 mg). Each of these three samples was divided into two equal subsamples allowing replication of all experiments.

## DNA extraction, library preparation and sequencing

To minimize contamination, all molecular work (pre-library amplification) was conducted in dedicated ancient DNA (aDNA) cleanlab facilities at the Lundbeck Centre for GeoGenetics, University of Copenhagen. The samples were first washed in molecular grade $H_2O$ to remove residual ethanol and then finely divided into smaller bits to facilitate a more efficient digestion. Samples were initially incubated for 48 hours at 42˚C on a rotator in 3.5mL digestion buffer, containing 3.25 mL 0.5 M EDTA buffer, 40 µL of Proteinase K (0.14–0.22 mg/mL, Roche, Basel, Switzerland), 175 µL 10% N-laurylsarcosyl, and 35 µL TE buffer (100x). After 24 hours the samples were not sufficiently digested and therefore an additional 40 µL of proteinase K was added to each sample. After further 24h incubation, the samples were centrifuged for 2 minutes at 2000×g and the supernatant was transferred to 50 mL tubes.

DNA extraction was done using silica-in-solution and a binding buffer that is efficient in recovering very short and degraded DNA fragments [44]. The buffer was prepared in bulk by mixing 500 mL buffer PB (Qiagen, Hilden, Germany) with 9 mL of sodium acetate (5 M), and 2.5 mL of sodium chloride (5 M), pH adjusted to 4–5 with 37% HCL. Supernatant from digestion was mixed with 40mL binding buffer, added 100µL silica suspension and incubated for one hour at room temperature. After DNA binding, samples were centrifuged at 2000×g for 5 minutes and the supernatant removed. The silica pellet with bound DNA was then resuspended in 1 mL of the same binding buffer, followed by two wash-steps with 1 mL 80% cold ethanol. Following 2 min of centrifugation at 2000×g, all the supernatant was removed and the pellets were left to dry in a laminar flow workbench (LAF) for 15 minutes. Finally, DNA from the washed and centrifuged silica pellets were eluted by adding 85 µL of EB buffer (Qiagen) supplemented with 0.05% Tween-20 detergent. After extraction, the DNA concentration (ng/µL) in the extracts was measured in two replicates for each sample by Qubit Fluorometric dsDNA high-sensitivity Quantification (ThermoFisher Scientific).

DNA extract was built into blunt-end libraries using the NEBNext DNA Sample Prep Master Mix Set E6070 (New England Biolabs, Ipswich, MA, USA) and Illumina-specific adapters (Illumina, San Diego, CA, USA). Protocol followed manufacturer guidelines with a few modifications as outlined in Orlando et al. [63] and Damgaard et al. [45], although the nebulization step was excluded since DNA in museum specimens must be expected to be sufficiently

fragmented already. The end-repair step was performed in 25.5 μL reactions using 21.25 μL of DNA extract and incubated for 20 mins at 12˚C and 15 min at 37˚C. Products were purified using Qiagen MinElute silica spin columns according to manufacturer guidelines, except that PB buffer was replaced with 250 μL of the modified binding buffer in the binding step. DNA was eluted in 15 μL Qiagen EB buffer and adapters were ligated to the blunt-end DNA in 25 μL reactions, using 5 μL of ligation buffer, 0.5 μL of adapter mix, 2.5 μL of Quick T4 ligase and 2 μL of $H_2O$. The reaction was incubated for 15 mins at 20˚C and products were purified with PB buffer on Qiagen MinElute columns, before being eluted in 23.5 μL EB Buffer. The adapter fill-in reaction was performed in 30 μL reactions using 23.5 μL of template, 3 μL of fill-in buffer, 2 μL dNTP mix (2.5 mM) and 1.5 μL of Bst polymerase and incubated for 20 mins at 37˚C followed by 20 mins at 80˚C to inactivate the enzyme. One μL of library was qPCR quantified with SYBRGreen on a Roche LightCycler 480 to determine the optimal number of PCR cycles required. Libraries were amplified and indexed in 50 μL PCR reactions, using 21 μL library template, 25 μL 1X KAPA HiFi HotStart Uracil + ReadyMix (KAPA Biosystems, Woburn, MA, USA) and 2 μL of both forward and reverse custom index primers. The reaction was amplified with an initial minute at 98˚C, followed by 7–14 cycles of 15 seconds at 98˚C, 30 seconds at 65˚C and 30 seconds at 72˚C, ending with one minute at 72˚C. The amplified libraries were purified using Agencourt AMPure XP beads (Beckman Coulter, Krefeld, Germany). DNA concentrations of the amplified libraries were quantified on an Agilent Bioanalyzer 2100 (Agilent Technologies) before making the pools for shotgun sequencing (80 bp, single read) on Illumina HiSeq 2500 at the Danish National High-throughput DNA Sequencing Centre.

## Bioinformatics and metagenomic analyses

Data were base-called using the Illumina software CASAVA 1.8.2 and sequences were de-multiplexed using a requirement of a full match of the six nucleotide indices that were used. The raw reads were trimmed for adapters using AdapterRemoval2 [64], and trimmed reads shorter than 30 bp were discarded. In order to assess the amount of endogenous DNA the data was mapped against the draft genome sequence of *T. sirtalis* [65]. The trimmed reads were mapped using BWA v. 0.7.15 aln [66] with default settings and duplicate reads were removed using the "rmdup" function in SAMtools v. 1.5 [67]. To investigate the level of DNA degradation in our samples, the mapped sequences were analysed with mapDamage v.2.0.5 [68] which quantifies DNA damage patterns, including base misincorporation levels and fragments lengths. These snake specimens are likely to have been handled many times over the years so as proxy for DNA contamination in the various tissues, we mapped the sequencing data against the human reference genome, build 37 (GRCh37) and recorded the fraction of mapped sequences for each sample.

To identify DNA from microbial organisms and thus gauge the metagenomic research potential in these old tissues, we used the taxonomic sequence classifier implemented in Kraken v2.0.7 [69]. We analyzed all our DNA sequences with default parameters and a confidence scoring threshold of 0.5 based on the standard Kraken database, containing all genomes of human, bacteria, fungi and archaea in the NCBI RefSeq database as of January 2019. The output of the Kraken2 classification was used as input for Bracken v2 [70] to obtain the abundance of taxa for each sample. The results of the metagenomic analysis were processed and visualized with the Pavian R package [71] and KronaTools v2.7 [72], respectively. We used the prcomp function in R to run a PCA analysis based on weighted number of hits obtained from the Kraken2/Bracken classification. The weighting was done by transforming the raw number of classified hits into fractions to account for differences between total number of DNA reads among

the samples. We used all thirty eigenvectors obtained from the PCA as input for the uniform manifold approximation and projection (UMAP) analysis using the "umap" R package. Venn diagrams were constructed using the "VennDiagram" 1.6 R package.

### Test for formalin exposure

An enzymatic test was performed to assess if the specimens had been exposed to formalin during preservation [61]. Two muscle samples of 0.5 x 0.5 x 0.5 cm were obtained from each specimen and suspended in 10% Savinase. Two parallel tests were performed (S1 Table), A) a 20 mL reaction being stirred in a beaker for 3 hours, and B) a 20 mL reaction in a red cap bottle with no stirring for 24 hours. According to the protocol, if the tissue has not degraded after 3 hours using stirring or 24 hours without stirring, it can be concluded that the specimen has been exposed to formalin [61, 73].

### Linear mixed effect models

Linear mixed effect models using maximum likelihood were used to analyze the effects of tissue matrix on yield of endogenous DNA, mapped length, CT damage, content of human DNA and DNA concentration. The models were constructed with specimen as random factor while tissue matrix was fixed factor. Bone was used as the reference matrix against which values for liver and skin were compared.

Another set of linear mixed effect models were used to investigate the effects of specimen age on the same independent variables. The models were constructed with specimen and tissue matrix as random factors while specimen age in years was fixed factor.

The raw values of both replicate samples from each specimen were used in the models. The variables human DNA content and DNA concentration were log-transformed to reduce skewness and approximate normal distribution. The analyses were performed using the R package *nmle* [74].

## Results

In general, we observed a high degree of consistency between results of replicate samples and we will therefore use average values in all the metrics discussed below, apart from the data substrate for the mixed effect model, which was the raw data. S2 Table shows results from all the DNA extracts individually.

### Formalin test

The results representing each of the two enzymatic treatments are presented in S1 Table. The specimen R602827 collected in 1923 was the only specimen that consistently showed clear evidence of exposure to formalin, whereas R602826 collected in 1920 was the only specimen that consistently showed clear evidence of no exposure. For the remaining three specimens, the results were inconclusive with the tissue being partly degraded (S1 Table).

### DNA concentration

Except for specimen R603729, the DNA concentration was highest in the liver tissue extracts, with four liver samples showing DNA concentrations between 22.7–92.4 ng/μL, whereas the oldest individual from 1842 (R602799) yielded only 5.7 ng/μL (Table 2). DNA concentrations from skin extracts were low in three samples (0.3–2.1 ng/μL) and relatively high in two samples (12.1 and 25.9 ng/μL). The DNA content in bone reflected similar values with two samples displaying low concentrations (1.8 and 2.6 ng/μL) and two samples with higher concentrations

**Table 2. DNA preservation.**

| Museum # | Collection year | Formalin | Tissue | Conc. (ng/μL) | Endo % | Length, bp | C-T % | Human % |
|---|---|---|---|---|---|---|---|---|
| ZMUC R602799 | 1842 | ? | Liver | 5.7 | 35.7 | 46.8 | 19.8 | 1.1 |
| | | | Bone | 2.6 | 42.2 | 51.5 | 14.5 | 0.5 |
| | | | Skin | 0.3 | 37.1 | 47.1 | 14.1 | 1.2 |
| ZMUC R602815 | 1880 | ? | Liver | 38.1 | 33.7 | 49.2 | 20.5 | 1.7 |
| | | | Bone | 10.5 | 47.2 | 51.5 | 16.0 | 0.6 |
| | | | Skin | 0.6 | 33.7 | 40.4 | 18.8 | 1.3 |
| ZMUC R602826 | 1920 | No | Liver | 92.4 | 47.4 | 59.4 | 11.1 | 0.3 |
| | | | Bone | 18.6 | 50.5 | 63.4 | 10.9 | 0.2 |
| | | | Skin | 25.9 | 46.8 | 59.3 | 9.1 | 0.2 |
| ZMUC R602827 | 1923 | Yes | Liver | 54.7 | 54.7 | 46.8 | 13.7 | 0.3 |
| | | | Bone | 1.8 | 55.6 | 49.7 | 16.8 | 0.3 |
| | | | Skin | 2.1 | 55.1 | 45.8 | 16.4 | 0.3 |
| ZMUC R603729 | 1964 | ? | Liver | 22.7 | 29.0 | 48.8 | 15.7 | 0.9 |
| | | | Bone | 83.3 | 48.5 | 62.1 | 8.8 | 0.2 |
| | | | Skin | 12.1 | 45.4 | 45.2 | 12.6 | 0.4 |

Results of formalin test, DNA concentration measurement and bioinformatic analyses. *Endo %*, endogenous DNA content; *Length*, average length (in base pairs) of mapped reads; *C-T %*, C-to-T deamination damage fraction recorded at first position in the 5'end; *Human %*, fraction of sequences that could be mapped to the human reference genome. Each value represents the average of two replicate DNA extracts (see S2 Table for all observations).

(10.5 and 18.6 ng/μL). For R603729, the DNA concentration in the bone extract was the highest (83.3 ng/μL). Overall, the oldest sample from 1842 (R602799) performed worst in regard to DNA concentration (Table 2, Fig 1). The linear mixed effects models revealed that tissue matrix had significant effects on DNA concentrations, and liver samples were estimated to provide the highest concentrations of DNA with 221% (p = 0,01) higher content than bone, while skin samples provided 73% (p = 0,004) lower DNA content than bone (S4 Table).

## Bioinformatic results

Approximately 570 million DNA sequences were generated for this project, ranging from 6 to 44 million, and averaging 19 million sequences per DNA extract (S2 Table). Following bioinformatic trimming and mapping against the *T. sirtalis* genome, between 29.0% (R603729, liver) and 55.6% (R602827, bone) of the sequences could be aligned (Table 2, Fig 1) thus representing the endogenous DNA content in these extracts. When clonal reads were removed, these values corresponded to an overall sequencing efficiency between 12.8% and 50.3% (S2 Table). For all five specimens, the endogenous DNA content was highest in bone tissue (Table 2, Fig 1), but in general the difference between tissue types was relatively modest. The linear mixed effects models, however, revealed that tissue matrix had significant effects on all qualitative metrics (Table 3). For endogenous DNA content, bone performed best with liver and skin estimated to produce 9 (p = 0.0008) and 5 (p = 0.0298) percentage points lower values, respectively (Table 3, Table 2, Fig 1). The difference in endogenous content between bone and liver obtained from each specimen ranged from 0.9% higher in bone in R602827 to 19.5% higher in bone in R603729 (Table 2, Fig 1). In four of the five specimens, skin and liver tissue samples showed almost equal levels of endogenous DNA content (Table 2, Fig 1). The average length of the mapped sequences was short and ranged from 40.4 to 63.4 bp per sample (Table 2, Fig 1). Notably, for all five specimens, bone also performed best, with liver and skin estimated to produce 5 (p = 0.003) and 8 bp (p = 0.0001) shorter fragments, respectively

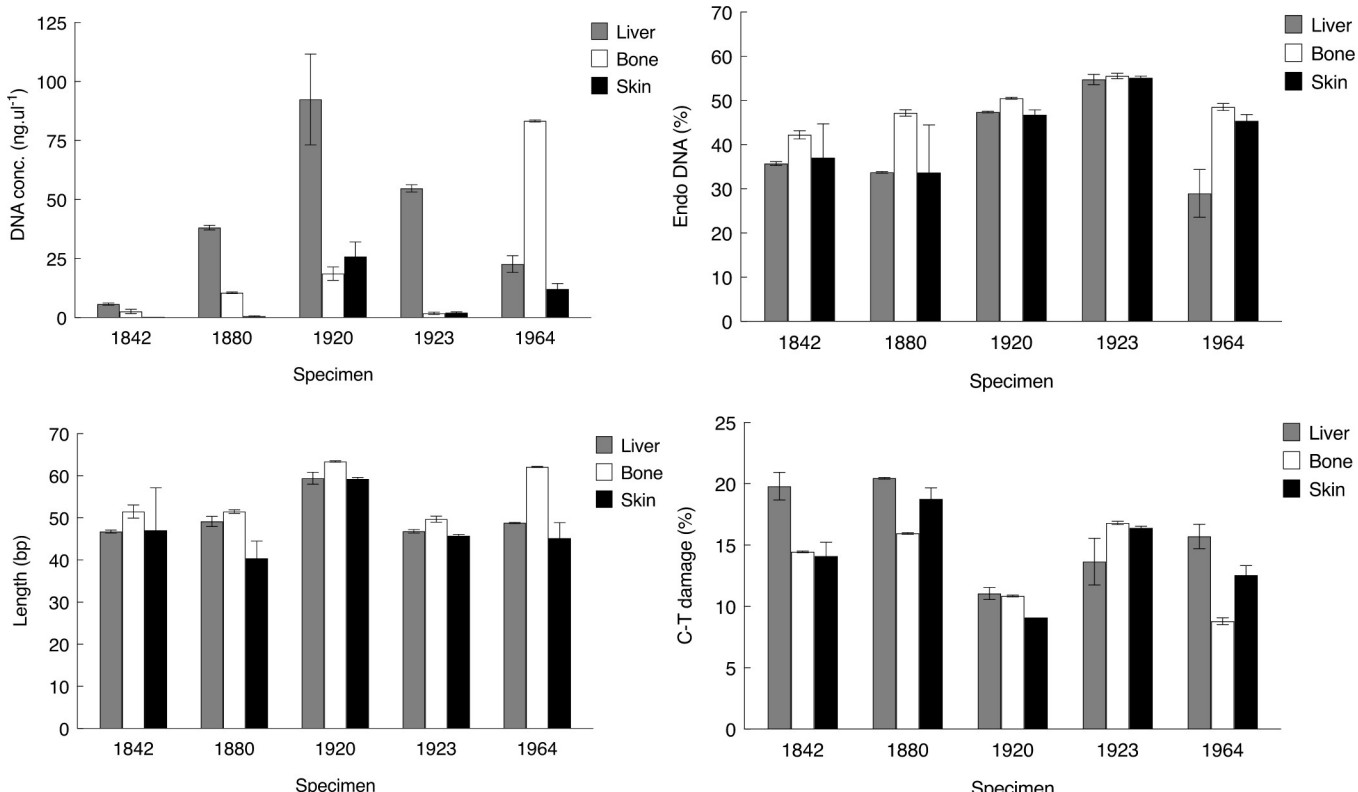

**Fig 1. DNA preservation.** DNA concentration measurement and DNA preservation metrics listed according to specimen age. *Endo %*, endogenous DNA content; *Length*, average length (in base pairs) of mapped reads; *C-T %*, C-to-T deamination damage fraction recorded at first position in the 5'end. Bars represent average values of two replicate DNA extracts with the range shown.

(Table 3). As with the endogenous DNA content, the sequence lengths in liver and skin were somewhat similar in length although these differences were significant (Table 3). We observed a high C-T deamination fraction, ranging from 9.1% to 20.5% at the first 5' position of the sequences. Bone was estimated to perform best with regard to CT damage, although the difference relative to skin was not statistically significant (Table 3). Samples from liver tissue were estimated to contain 2.76 percentage points (p = 0.0057) more damage than bone, while skin samples were estimated at 0.81 percentage points (p = 0.38) higher (Table 3). In summary, we see a relatively high level of endogenous DNA content in these samples, but the data reflect short and damaged DNA molecules.

Overall, we found no obvious trend between the different measures of DNA preservation and the age of the specimen (Figs 1 and 2), apart from observing a very low DNA concentration in the extract of the oldest one from 1842 (R602799). The linear mixed effects model on sample age revealed that in our samples, age had very modest and statistically insignificant effects on all qualitative metrics (S4 Table).

Only two specimens could be assigned with certainty with regard to the presence/absence of formalin, but since these two specimens have almost the same age (1920 and 1923), it allows for a direct comparison of the potential effects of formalin. However, due to the small sample size, these comparisons should only be observed as potential indications. The specimen without formalin exposure (R602826) displayed higher DNA concentration, longer average mapped read lengths, as well as less C-T damage, in all tissues, in comparison to the specimen exposed to formalin (R602827). The non-formalin exposed specimen also showed the longest

**Table 3. Linear mixed effects model.**

| Endogenous DNA, % | | | | | |
|---|---|---|---|---|---|
| Source | Effect | SE | df | t | p |
| Intercept | 48.79 | 3.54 | 23 | 13.79 | <0.0001 |
| Liver | -8.69 | 2.24 | 23 | -3.88 | 0.0008 |
| Skin | -5.19 | 2.24 | 23 | -2.34 | 0.0298 |
| **Mapped length, bp** | | | | | |
| Source | Effect | SE | df | t | p |
| Intercept | 55.64 | 2.73 | 23 | 20.42 | <0.0001 |
| Liver | -5.46 | 1.66 | 23 | -3.28 | 0.0033 |
| Skin | -8.11 | 1.66 | 23 | -4.87 | 0.0001 |
| **CT damage, %** | | | | | |
| Source | Effect | SE | df | t | p |
| Intercept | 13.37 | 1.52 | 23 | 8.78 | <0.0001 |
| Liver | 2.76 | 0.91 | 23 | 3.05 | 0.0057 |
| Skin | 0.81 | 0.91 | 23 | 0.89 | 0.3804 |
| **Human DNA, %** | | | | | |
| Source | Effect | SE | df | t | p |
| Intercept | -67% | 139 | 23 | -3.32 | <0.0001 |
| Liver | 89% | 113 | 23 | 5.15 | <0.0001 |
| Skin | 50% | 113 | 23 | 3.28 | 0.0033 |
| **DNA conc, ng µl⁻¹** | | | | | |
| Source | Effect | SE | df | t | p |
| Intercept | 837% | 185 | 23 | 3.63 | <0.0001 |
| Liver | 221% | 152 | 23 | 2,79 | 0.0103 |
| Skin | -73% | 152 | 23 | -3.16 | 0.0043 |

Results of the linear mixed effects models conducted on endogenous DNA content, mapped length, CT damage, human DNA content and DNA concentration. Bone was used as the reference matrix against which values for liver and skin were compared. The estimated effects derived from the log-transformed variables (human DNA content and DNA concentration) are given in percentage deviations from bone, while other effects are given in the relevant units (percentage points or base pair length). SE: standard error, df: degrees of freedom, t: test value, p: p value.

average read length of all five samples. Conversely, the specimen with confirmed formalin exposure showed the highest endogenous DNA content of all the five specimens.

These snake specimens are likely to have been handled many times over the years so as proxy for DNA contamination in the various tissues, we mapped the sequencing data against the human reference genome. Based on this, we could estimate the proportion of DNA sequences with a human origin. Overall, the human DNA contamination levels were low, ranging from 0.2% to 1.7% of the reads (Table 2, Fig 2). Across individual tissue types, the average human contamination was 0.8% in the liver samples, 0.5% in the skin samples and 0.3% in bone tissue. The two oldest samples (R602799 and R602815) had the highest average contamination fraction of 0.9% and 1.2% respectively, whereas the remaining samples had average contamination between 0.2% and 0.5%. Despite these low values, we observe that DNA contamination levels across different tissue types are somewhat correlated when they represent the same specimen (Fig 2). Thus, if a specimen displays an increased level of human DNA contamination, this is likely the case for all three types of tissue. The linear mixed effects models also revealed that bone was superior with regard to human DNA content, where liver

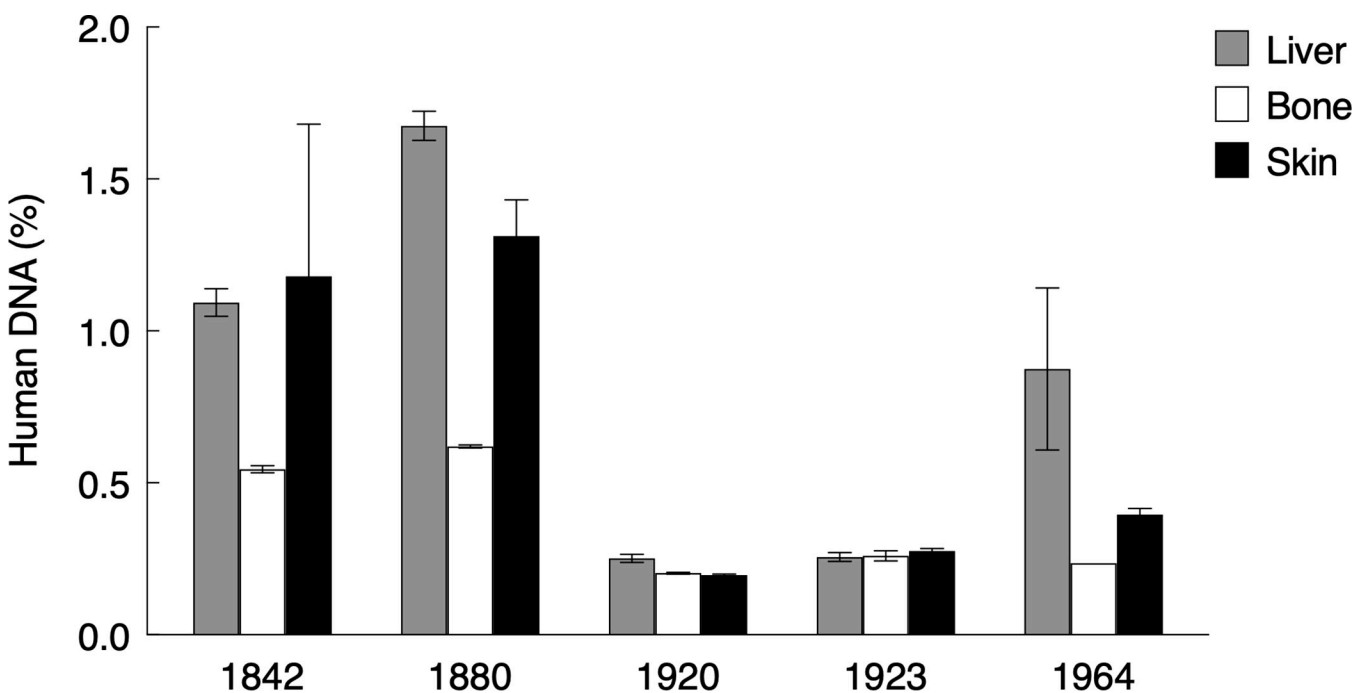

**Fig 2. Human DNA contamination.** Fraction of sequences that could be mapped to the human reference genome listed according to specimen age. Bars represent average values of two replicate DNA extracts and the range is shown.

and skin, despite the low absolute values in all matrices, were estimated to contain 89% (p = <0.0001) and 50% (p = 0.0033) more contamination respectively (Table 3).

In the metagenomic analyses, we binned the sequences according to tissue types but combined them across individuals. We applied a species-level designation but to reduce background 'noise' we considered only taxa with >5 sequence hits against the reference database. It is beyond the scope of our study to provide an accurate and complete taxonomic characterization of the microbial community in these samples. We focused on the overall metagenomic differences in exploring the potential and we note that exact taxon matches should be interpreted with caution. Nonetheless, the Kraken2 analyses showed clear differences in diversity between the tissues (S3 Table), most clearly observed in the bacterial community (Fig 3) where 441 taxa are observed in the skin samples, 309 taxa in the bone samples, and 289 taxa in the liver samples. A considerable taxonomic overlap between tissue types is visualized in a Venn diagram (Fig 3), and it is also clear that a substantial part of the diversity is observed in the extraction blanks and thus represent DNA introduced with reagents or during sequencing despite working in clean-labs, and despite observing unmeasurable levels of DNA concentration in these negative extraction controls. Taking a conservative approach, by simply ignoring all taxa that are also observed in the extraction blanks, we still observe 106 different bacterial taxa in our samples, 30 of which are exclusively observed in the skin samples.

We applied a principal component analysis (PCA), as well as adding further dimensionality using a Uniform Manifold Approximation and Projection (UMAP), in order to assess the relationship between the metagenomic profiles from all tissues and individuals. In both the PCA and most clearly observed in the UMAP results, we observe some clustering according to individuals (S1 Fig) and in the PCA analyses, the skin/muscle tissue displays the highest apparent metagenomic variation. Moreover, in the PCA we also see some clustering by tissue types (S1

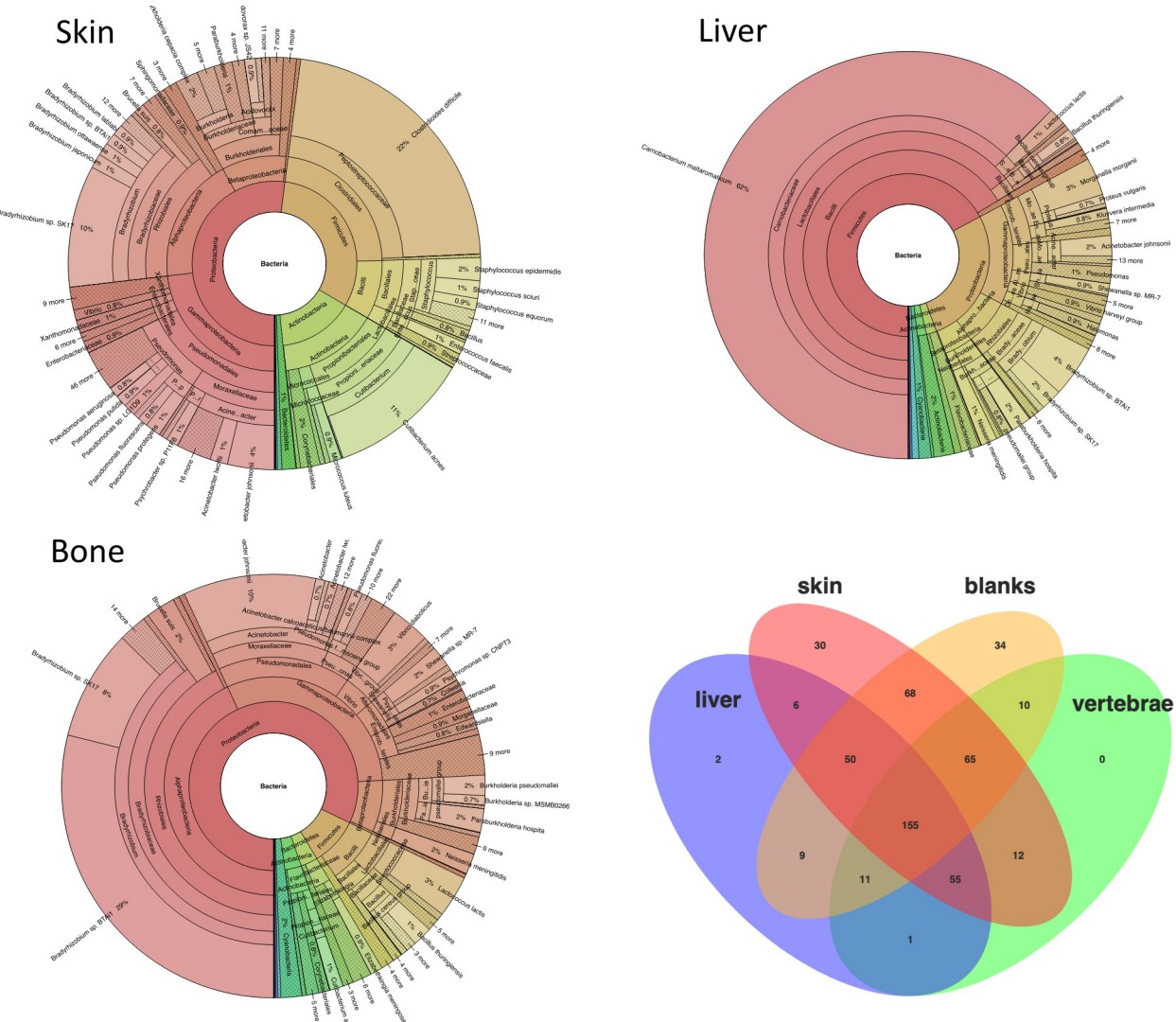

**Fig 3. Metagenomic profiling.** Bacterial metagenomic diversity visualized in Krona and Venn (lower right) plots for the three tissue types. The diversity is presented at species-level and includes only NCBI refseq taxa with >5 sequence matches. Colors denote taxonomic groups found within the various bacterial phyla as according to Krona visualization. *Blanks* denote negative extraction control.

Fig), but since there was much variation between extracts from the same samples, this indication should be interpreted with caution.

## Discussion

Owing to NGS technology, genome sequencing has become an affordable and widely used element in many areas of biological research. Yet, the number of studies that thoroughly explore the genomic potential of old museum specimens is still limited. By using an ancient DNA methodology, we show that genomic analyses of old ethanol-preserved museum material are indeed feasible, displaying great promise for unlocking the wealth of molecular information stored in the millions of specimens found in museum collections around the world. The successful generation of NGS data from museum specimens is not unique [34–37, 75, 76], but very few studies have systematically tested different tissue types with the aim of establishing

best practice protocols when sampling valuable museum material for genomic research. Below we will discuss the various DNA preservation metrics included in our study.

## DNA concentration

The DNA concentration serves as a basic quality measure in many molecular contexts, because it reveals if the extraction has been successful and it can inform on how the DNA should be prepared in downstream experimental work. It is, however, only a crude indicator of the usability in genome-scale sequencing because DNA concentration can be adjusted post-extraction, and most library methods require only very limited input DNA anyway–in particular methods that have been optimized for aDNA research as in the current setup. Regardless, DNA could be detected in all our extracts showing that the extraction method has been successful despite using old ethanol-preserved specimens. This includes the formalin-exposed specimen, where we observed high DNA content (53.6–55.8 ng/μL) in the liver tissue extracts. Except for sample R603729, the DNA concentration was highest in liver tissue, ranging from 22.7–92.4 ng/μL. Similar results were found by Hykin et al. [33], comparing DNA concentration in liver, muscle and tail tips from two specimens of *Anolis caroliensis* collected approximately 30 years ago.

We generally observed lower DNA concentration (0.3–26.9 ng/μL) in skin extracts. Since skin, including also snake scales, are partly made of dead keratinized cells [77], it is not unexpected that these extracts contain lower amounts of DNA than liver tissue. Similar results were also obtained by Casas-Marce et al. [78] who compared DNA contents in skin versus bone samples from museum Iberian lynx (*Lynx pardinus*) as well as Tsai et al. [32] who found the lowest DNA concentration from dried skin tissue compared to bone and toe-pad tissues. Moreover, skin, in comparison with bone and liver tissue, is also in direct contact with the environment, which could potentially elevate processes leading to DNA damage.

The linear mixed effects model showed significant differences in DNA concentrations from the various tissues, but as discussed above this is not a particularly informative metric when assessing the potential for genomic-scale analyses.

## Endogenous DNA

The endogenous DNA content is the most important factor when working with degraded samples in a genomic context, because it defines how much 'shotgun' sequencing is required to reach a certain genomic coverage. This is particularly pertinent in the aDNA field since most biological tissues exposed to natural environments for centuries or millennia will display mostly DNA of a microbial origin and often less than 1% endogenous DNA [41, 79, 80]. Therefore, a considerable effort in aDNA research has been aimed at maximizing the endogenous DNA content through improved sampling, DNA extraction, and library preparation protocols [42–45, 49, 81].

By mapping our many millions of DNA sequences to the *T. sirtalis* reference genome we found a relatively high fraction of endogenous DNA in all the individuals and tissues, ranging from 26% to 56%. Although this is still markedly lower compared to working with fresh tissue, where the endogenous DNA fraction is expected to be close to 100%, we argue that all tissues from the included specimens have adequate levels of endogenous DNA for genome-wide sequencing to be feasible. In support of the sufficiency of these levels for whole genomic sequencing, whole genomes (>1X depth of coverage) are routinely sequenced from endogenous content of similar values or lower, down to c.10% in the ancient DNA field [44, 82–85]

The relatively high endogenous DNA content in our extracts shows that long-term storage in 70% ethanol does not preclude an effective shotgun sequencing approach and genome-scale

research. Presumably this type of storage will limit the introduction of fresh/living microbes to the specimen, thereby maintaining a relatively high endogenous DNA fraction over decades and even centuries.

Many studies have reported the successful extraction and sequencing of DNA from old museum specimens (see Burrell et al. [12], Wandeler et al. [14], Holmes et al. [19]), but very few have systematically tested different samples and tissues for their endogenous DNA content based on millions of reads. Exceptions are human skeletal remains that have been extensively tested in recent years in the aDNA field. Concerning non-human taxa, endogenous DNA fractions from various tissues have been assessed in Great Apes from museum collections [86], displaying fractions between 1–99% in teeth and >80% in soft tissue samples (fingertips and ear cartilage). Similar variation is evident from other museum collections samples. For example, the genomic mapping rate to reference genomes was c. 40% in dried Alpine chipmunk *(Tamias alpinus)* skin samples [35] and between 21.8–38% from rat (*Rattus norvegicus*) molar and toe samples [34]. Likewise, Allentoft et al. [13] used an aDNA framework to extract and shotgun-sequence samples from ethanol-preserved Aesculapian snake tissues (muscle/skin, teeth and vertebrae), for phylogenetic purposes, but also tested the mapping rate against the *T. sirtalis* genome. Without a suitable reference genome, the mapping rate was much lower (0.6–8.6) than we observe here, but all tissue types proved positive for snake DNA.

The analyses showed that the differences in endogenous DNA between all three tissues were significant (Table 3). However, when evaluating which of our included substrates performed best, it should be acknowledged that liver and skin showed almost equal levels of endogenous DNA in all samples in absolute values, except in extracts from the most recently collected individual from 1964 (R603729), where skin was 16.4 percentage points higher than liver tissue. In all five specimens, bone tissue contained the highest fraction of endogenous DNA. Bone can act as a barrier against autolytic, oxidative, and hydrolytic damage [87], and perhaps also limit DNA degradation caused by unfavorable pH conditions [88] therefore more intact and non-damaged molecules may be available from this substrate. Also, bone is a denser tissue than liver and skin, and therefore presumably less permeable to invading microbes *post-mortem*. These factors arguably explain why bone displays the highest endogenous DNA content in our study. Likewise, in the above-mentioned rat study, Rowe et al. [34] obtained the longest DNA fragment lengths and highest total nucleotide concentrations from bone samples. Similarly, Allentoft et al. [13] obtained the highest mapping efficiency from bone tissue, compared to muscle/skin and teeth. Moreover, previous studies on squamate reptiles from museum collections also succeeded in extracting and sequencing nuclear DNA from both skeletonized bone and bone preserved in 70% ethanol and presumably also exposed to formalin [89, 90].

However, it is important to note that the absolute values we have obtained from bone are only slightly higher than those from the other tissues (Table 3). For example, the average endogenous DNA content for bone was 50.4%, whereas the average was 45.2% and 41.2% for skin and liver respectively. This is relevant for sampling strategy purposes, since sampling of skin and liver tissue is generally less complicated and invasive than sampling bone from an intact specimen preserved in ethanol. In many instances, it may therefore be advisable to sample skin or liver, in particular when sampling rare and valuable specimens, such as type specimens or extinct species where materials are rare.

Interestingly, we observed the highest endogenous DNA content (evident in all three tissues), in the specimen that proved positive for formalin exposure. Low-coverage genomic DNA has previously been reported from formalin fixed museum specimens [30, 33], although these studies encountered mixed DNA extraction success rates and DNA quantities. Ruane and Austin [30] were able to sequence Ultra-Conserved Elements (UCEs) from a c. 100 years

old fluid and formalin preserved snake, showing that it is indeed possible to obtain useful data from a century old formalin-exposed sample. In this study, however, we mapped sequences to a reference genome to obtain a measure of the actual endogenous DNA content, and thus we are able to document the genomic research potential in an old formalin preserved snake from 1923 (R602827). It is perhaps not entirely surprising that we find a relatively high level of endogenous DNA in a formalin fixed sample. Formalin fixation infers several crucial alterations that complicates DNA extractions and can also elevate DNA fragmentation rates [27]. These processes can ultimately reduce DNA concentration, increase deamination damage or reduce fragment length, but we do not have reason to expect that formalin will lower the endogenous DNA content–perhaps rather the opposite as formalin exposure is likely to reduce microbial attack and decomposition. It is well known that formalin fixation may hamper or completely prevent successful recovery of DNA from a given sample, but our results show that if DNA extraction is successful, the quality and purity (i.e. endogenous DNA content) of the recovered DNA is just as good–and perhaps even better–than what is observed in non-formalin exposed samples of similar age.

## DNA damage and contamination

We also investigated other proxies for DNA quality, namely average fragment lengths and levels of C-to-T deamination damage at the first 5' position of the sequences [91, 92]. In a normal metabolically active cell, the DNA molecules are rapidly and efficiently repaired by the hosts repair pathway, but *post-mortem* errors in the DNA accumulate, resulting in fragmentation and C-to-T misincorporation [20, 21]. The rate at which this happens depends on microenvironmental factors such as temperature, pH and salinity [93–95].

Overall, we observe high C-to-T damage (9.1% to 20.5%) in all samples. These values are surprisingly high, reflecting severely damaged DNA, equivalent to what is often observed in bones that are thousands of years old (e.g., Allentoft et al. [44]). In all except one specimen (R603729) the C-T deamination was highest in liver tissue (11.1–20.5%), whereas bone had the lowest average deamination across tissue type (13.1%) but when assessing the absolute values the differences are subtle. The specimen from 1920 (R602826), from where formalin exposure could not be detected, had the lowest average C-T values (10.3%) across tissues. C-T deamination has been reported in countless aDNA studies but is less commonly reported in studies using more recent (i.e. decades, centuries) museum material. When investigating C-T deamination in a timeseries of monkey/gorilla specimens, Sawyer et al. [93] reported rates <4% for the majority of samples younger than 120 years. Only specimens that were burned or more than 500 years old showed C-T deamination rates >9.1% (our lowest value). Similar values have been obtained from herbarium museum samples covering the last 300 years, giving values ranging from c. 1,5–5% [96]. In contrast, a recent study by Allentoft et al. [13] found 11–15% C-T deamination in old (1810 to 1863) ethanol preserved Aesculapian snakes. These values are similar to what we observe here, indicating that ethanol-preservation (possibly combined with exposure to formalin) results in a higher damage rate than in dry museum material. C-T damage is not necessarily a major problem in downstream analyses, and it can be used to authenticate that the DNA is indeed of ancient or historical origin. However, a high level of C-T damage could result in a lower reference genome mapping affinity in the bioinformatical analyses, and hence the loss of relevant data. Moreover, if not accounted for analytically, C-T misincorporation could be wrongly interpreted as true genetic variation, resulting in problematic conclusions.

Similarly, the average DNA fragment lengths we observe (40–64 bp), also reflect highly degraded DNA. The longest average lengths were found in specimen R602826 (59.3–63.4 bp),

where formalin exposure was not detected, but these values are only marginally larger in absolute values than in the other specimens. However, we will emphasize that such cross-study comparisons of fragment lengths should be carried out with caution as they can be influenced by the extraction method, the library method, the PCR and sequencing chemistry. Still, it is clear that these values represent highly degraded DNA, similar to what is often observed in ancient biological substrates. Such highly fragmented DNA also explains why PCR-based methods often fail with museum material [27, 28, 97] since there is simply not enough space for binding primers to the template DNA.

Following the discussion above, it seems that storage in 70% ethanol at room temperature is not ideal for long term DNA preservation. A reason could be that in a 70% ethanol solution (as widely used in museum collections) the presence of 30% water will promote spontaneous hydrolysis of the DNA [21, 98]. Long-term storage will likely lead to lowered ethanol concentration due to evaporation, and possibly also unfavorable pH conditions [88] eventually increasing the rate of DNA degradation further [27].

Because DNA degrades *post-mortem*, we would intuitively expect that the DNA quality would decline with specimen age. Previous results on museum animal remains [93] and herbarium samples [96] found strong correlation between C-T deamination and time. However, a correlation between average fragment lengths and time was only observed from the herbarium samples [96]. The oldest snake in our study had the lowest DNA concentration in all three tissues, but aside from this observation, we found no correlation between age and the measured variables in the linear mixed effects model on sample age (S4 Table). Similar lack of correlation between age and state of DNA preservation was recently also observed from museum samples covering c. 100 years [99], including both ethanol preserved and dried samples.

As a final measure of DNA purity, we examined the contamination levels. It is perceivable that many years of storage in museum collections combined with periodic handling of the specimen could result in human DNA contamination. We found low levels of human contamination (0.2–1.7%) in this study, and although the differences among tissues are significant (Table 3), the levels of the observed absolute differences suggest that this is not a major concern in our case. Previous studies based on historical museum material have identified similar low levels of human DNA contamination. Using the human mitochondrial genome as reference the fraction of contamination was estimated to c. 1% in [86] and 2,3% in [36] and when based on genome-wide shotgun data, contaminant fractions of 4.3–8.9% in [100], <0,3% in [35], <0,1% in [34], and in 0.27% [33] have been reported. In summary, human DNA contamination has not severely hampered the genomic potential, and in most cases, a small proportion of human DNA sequences will not affect downstream analyses as they can easily be filtered out bioinformatically. Cross-contamination between specimens preserved in the same jars for many years might present a larger problem, but to our knowledge this remains to be investigated.

## Metagenomic potential

In order to assess the metagenomic profiles of these specimen and tissues, we examined the many shotgun-sequences that could not be identified as snake DNA. From the entire dataset, the bacterial DNA fraction accounted for just 0.01–0.02% of all sequences in bone and liver respectively (Fig 3), whereas the fraction in skin was higher at 0.06%. This seem to reflect a very low concentration of microbial DNA compared to snake DNA in the tested tissues and this is probably explained by preservation in ethanol which prevents *post-mortem* microbial growth. According to our knowledge, no previous study has performed shotgun metagenomics on ethanol-preserved museum specimens, limiting comparative assessments. A

different explanation for the low fractions could also be that we simply do not have sufficient database coverage of the microbial taxa found on snakes. Only few studies have investigated snake metagenomics [101–105] and these have all analyzed microbes from the digestive tract in fresh tissues, not from ethanol preserved museum specimens, and none of them have utilized shotgun sequencing. Thus a large fraction of snake-related bacterial genomes are likely to be absent from the databases [106]. Similar low fractions of reads being assigned to taxa have also been observed in aDNA studies [107–109], and although these results are not directly comparable to those obtained from ethanol preserved specimens, they still underline missingness in reference databases.

If the low fraction of reads assigned to known microbial diversity is partly a consequence of preservation in ethanol preventing *post-mortem* microbial growth, it could be expected that the microbial DNA profiles reflect the diversity as it appeared in the snakes at the time of preservation. The fact that we observe tissue-specific metagenomic profiles (Fig 3), and to some extent also specimen-specific ones (S1 Fig), suggest that this is indeed the case.

Under the strict criteria applied we found 30 bacterial taxa uniquely present in the skin tissue, whereas we only found two taxa uniquely associated with liver tissue, and no taxa uniquely found in bone. Because the skin of the animals has more direct contact with the environment than the other tissues, this result is perhaps not surprising but again, it is indicative that these tissues have retained characteristics expected for their original microbial profiles. However, we cannot rule out that some proportion of the microbial diversity observed in the skin is derived from the ethanol used for preservation.

Our metagenomic profiling is intended as a proof-of-concept to evaluate the potential and we caution that the exact taxonomic identification of specific microbial taxa require rigorous follow-up investigations. Nonetheless, we highlight here a few examples of the microorganisms most frequently observed in the snake specimens (S3 Table). *Edwardsiella tarda* was observed in two of the snakes, with highest prevalence in the skin tissue. This bacterium is known to infect fish, amphibians, mammals and reptiles and causes Edwardsiella septicemia also known as fish gangrene [110]. *Enterococcus faecalis*, observed in several of the snakes but predominantly in skin tissue, form part of the commensal gastrointestinal microbiota of a diverse range of taxa, including mammals, birds, reptiles and insects [111]. Bacteria in this genus are fast-emerging as both human and reptile pathogens and have previously been connected with diseases in sea turtles [112]. *Morganella morganii* was observed in all tissues from one of the individuals (R603729) but with the highest levels in skin. This bacterium is commonly found in the environment and intestinal tracts of humans, mammals and reptiles as part of the normal flora [113] and has been found as the primary agent of secondary wound infections after snakebites [114–116]. Interestingly the taxon displaying highest overall levels in all samples and tissues, covering 97–100% of all virus sequences, is Taterapox virus; which is a virus phylogenetically very closely related to *variola*—the etiological agent of smallpox. Only few studies have examined the genomes from this virus, and besides indications of the virus infecting Mongolian gerbils (*Meriones unguiculatus*) and mice [117] we cannot currently explain why this virus is present in all our samples, but it is an interesting observation that warrants further research.

Our results suggest that there is indeed authentic metagenomic information stored alongside the actual ethanol-preserved museum specimen. This could serve as a window into past microbial communities related to for example, infections and diet (i.e. microbiome), or to investigate the prevalence of certain pathogens through space and time. Because museum specimens always have a collection date and locality associated with them, these data also present an opportunity to directly assess the micro-evolution of historical bacteria and viruses. Thereby the potential benefit of applying metagenomic analyses similar to ours, or microbial

metabarcoding approaches, could add another highly informative layer to the many possibilities from museum samples, that were previously not possible in the pre-genomic era.

## Conclusion

By using an ancient DNA methodology, we were able to extract and analyse both genomic and microbial DNA from all these ethanol-preserved snakes, including a 178-year-old specimen and one exposed to formalin. Bone samples performed better by the applied quality standards (endogenous DNA content, DNA damage and contamination) and therefore, if we were to recommend a tissue type for genomic analyses, based solely on these values, bone would be preferable. However, although the observed differences between tissues were statistically significant, the absolute values were marginal. Given the difficulty and damaging nature of sampling a bone from a complete ethanol-preserved museum specimen, we will not, based on our results, advocate selective sampling of bone in favour of other tissues.

Importantly, all our samples displayed a relatively high endogenous DNA content, which confirms the genomic research potential from these museum specimens. We find that high endogenous content in combination with the aDNA and shotgun sequencing setup employed here, seems like a very advantageous setup for museum sample genomics. However, the endogenous DNA levels are still lower, and the DNA is much more fragmented than observed in fresh samples. Consequently, obtaining a certain level of genomic coverage will require a lot more sequencing compared to sequencing fresh samples, but with the values we have demonstrated here, generating useful genome-wide data is indeed possible.

Although our results are encouraging, further improvements could be explored. In all our samples, we found high levels of C-to-T deamination which may compromise downstream bioinformatic analyses. One way to lower these "error rates", could be to utilize the Uracil-Specific Excision Reagent (USER enzyme), which removes uracil residues, as widely applied in aDNA research [118]. Moreover, single-stranded library preparation protocols have been shown to significantly improve the output from ancient samples [119, 120], fragmented samples [121] and tissues stored in formalin [122]. Applying these methods to samples from ethanol-preserved specimens, similar to those we have tested, could prove beneficial and increase the research potential even further. It is important to keep optimizing and testing these methods, because museum specimens represent a valuable and finite resource and when performing destructive sampling for genetic analyses, it is imperative that everything possible is done to maximize the output.

## Supporting information

**S1 Fig. PCA/UMAP.** PCA analysis based on weighted number of hits obtained from the Kraken2/Bracken classification for each extract. The weighting was done by transforming the raw number of classified hits into fractions to account for differences between total number of DNA reads among the samples. The different types of tissue are noted. Uniform Manifold Approximation and Projection (UMAP), was used to add further dimensionality in order to assess the relationship between the metagenomic profiles from all tissues and individuals. Blanks were excluded for the analysis.
(PDF)

**S1 Table. Complete formalin results.** Results of formalin test using two test methods.
(DOCX)

**S2 Table. Complete data overview for all extracts.**
(XLSX)

**S3 Table. Kraken2/Bracken output.** Data was classified with Kraken2 and used as input for Bracken to obtain the abundance of all taxa for each sample. Total counts of tissue-specific taxa across all samples are shown to the right. Colors highlight the tissue type used for DNA extraction in the different extracts and blanks (blue = liver, green = vertebrate/bone, red = skin/muscle, orange = blanks).
(XLSX)

**S4 Table. Mixed models on sample age.** Linear mixed effect models were used to investigate the effects of specimen age on the same independent variables. The models were constructed with specimen and tissue matrix as random factors while specimen age in years was fixed factor.
(XLSX)

## Acknowledgments

We thank Jesper Stenderup and the staff at the Danish National High-throughput Sequencing Centre for technical assistance. We thank the Natural History Museum Denmark for providing access to their collections and allowing sampling of the specimens.

## Author Contributions

**Conceptualization:** Peter Gravlund, Arne R. Rasmussen, Morten E. Allentoft.

**Formal analysis:** Claus M. Zacho, Martina A. Bager, Anders Galatius, Morten E. Allentoft.

**Funding acquisition:** Morten E. Allentoft.

**Investigation:** Claus M. Zacho, Martina A. Bager, Arne R. Rasmussen, Morten E. Allentoft.

**Methodology:** Claus M. Zacho, Martina A. Bager, Peter Gravlund, Anders Galatius, Arne R. Rasmussen, Morten E. Allentoft.

**Project administration:** Peter Gravlund, Arne R. Rasmussen, Morten E. Allentoft.

**Resources:** Ashot Margaryan, Anders Galatius, Arne R. Rasmussen, Morten E. Allentoft.

**Software:** Ashot Margaryan, Anders Galatius.

**Supervision:** Ashot Margaryan, Peter Gravlund, Anders Galatius, Arne R. Rasmussen, Morten E. Allentoft.

**Validation:** Ashot Margaryan, Anders Galatius, Morten E. Allentoft.

**Visualization:** Claus M. Zacho, Martina A. Bager, Ashot Margaryan.

**Writing – original draft:** Claus M. Zacho, Martina A. Bager, Morten E. Allentoft.

**Writing – review & editing:** Claus M. Zacho, Martina A. Bager, Peter Gravlund, Anders Galatius, Arne R. Rasmussen, Morten E. Allentoft.

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
