## [Decision Letter · Decision Letter 0]

8 Apr 2021

PONE-D-21-05242

Uncovering the genomic and metagenomic research potential in old ethanol-preserved snakes

PLOS ONE

Dear Dr. Zacho,

Thank you for submitting your manuscript to PLOS ONE. After careful consideration, we feel that it has merit but does not fully meet PLOS ONE’s publication criteria as it currently stands. Therefore, we invite you to submit a revised version of the manuscript that addresses the points raised during the review process.

Please see the reviewer comments provided; there are some concerns regarding the small sample size and lack of statistical comparisons. In addition, the applicability of the findings is unclear for applications such as population and conservation genetics, as well as metagenomics given the potential issues with the DNA extracted. I think that most of these concerns can be addressed with careful revisions to the text, and potentially some additional analyses. I ask that the authors consider and address all of the reviewer comments.

We look forward to receiving your revised manuscript.

Kind regards,

Christopher M. Somers

Academic Editor

PLOS ONE

Additional Editor Comments:

I have received detailed comments from 2 subject experts, and I have reviewed the manuscript myself. All three of us find the work to be meritorious and potentially useful; however, there are some concerns with the strength and scope of the work given the small sample size and lack of statistical support. I ask that the authors consider all of the comments raised by the reviewers, with a particular focus on those that are technical / analytical in nature.

Journal Requirements:

2. We note that you are reporting an analysis of a microarray, next-generation sequencing, or deep sequencing data set. PLOS requires that authors comply with field-specific standards for preparation, recording, and deposition of data in repositories appropriate to their field. Please upload these data to a stable, public repository (such as ArrayExpress, Gene Expression Omnibus (GEO), DNA Data Bank of Japan (DDBJ), NCBI GenBank, NCBI Sequence Read Archive, or EMBL Nucleotide Sequence Database (ENA)). In your revised cover letter, please provide the relevant accession numbers that may be used to access these data. For a full list of recommended repositories, see http://journals.plos.org/plosone/s/data-availability#loc-omics or http://journals.plos.org/plosone/s/data-availability#loc-sequencing.

Reviewers' comments:

Reviewer's Responses to Questions

**Comments to the Author**

1. Is the manuscript technically sound, and do the data support the conclusions?

Reviewer #1: Yes

Reviewer #2: Partly

2. Has the statistical analysis been performed appropriately and rigorously? 

Reviewer #1: No

Reviewer #2: N/A

3. Have the authors made all data underlying the findings in their manuscript fully available?

Reviewer #1: Yes

Reviewer #2: Yes

4. Is the manuscript presented in an intelligible fashion and written in standard English?

Reviewer #1: Yes

Reviewer #2: Yes

5. Review Comments to the Author

Reviewer #1: Zacho et al. PLOS ONE

Uncovering the genomic and metagenomic research potential in old ethanol-preserved snakes

This manuscript examines DNA degradation in old museum preserved samples. They sample three different tissue types, skin, bone and liver, from five common garter snake (Thamnophis sirtalis) samples. These samples have been collected in different regions and different times, which allows them to examine different preservation techniques and the impacts of sample age on degradation. The authors use ancient DNA protocols and shotgun sequencing to determine the best collection, extraction and sequencing procedures for the analysis of museum specimens. Multiple different metrics, including DNA concentration, percent endogenous DNA, fragment length, C-to-T deamination damage and percent human contamination, are used and the authors find similar results across tissue types. In general, I found this manuscript to be of interest and a valuable contribution to the field. However, I did have several concerns with the manuscript that should be addressed.

Major Comments:

1. The main question of the manuscript was to determine the best approach for sequencing museum specimens and compared multiple different tissues. I believe that these analyses should be more rigorous and should include statistical analyses. The authors compare multiple metrics (Figs. 1 and 2) across tissues and specimens and the addition of statistical comparisons through an ANOVA or a similar analysis would greatly improve the manuscript. This would allow the authors to statistically compare the different tissues within a sample. Further, I believe that statistically testing for differences across samples (ie. comparing liver samples across all specimens) would also strengthen the paper. This would allow the authors to compare the ages of the samples as well as the different storage and preparation techniques used by museums.

2. Further to the comment above, the authors should include standard deviations in the manuscript when they are using averages.

3. The manuscript is relatively well written, but I believe there are 3 areas that would help improve the clarity of the study:

a. The introduction is quite clear but the last 3 paragraphs need to be more concise. These paragraphs could be combined into 2 paragraphs that would be clearer. The first paragraph should address the similarities to aDNA and the availability of the metagenomic profile. The second paragraph should clearly introduce the objectives and methods of the study.

b. The discussion is quite long and needs to be shortened for clarity. Each of the sections within the discussion should be clearer and also clearly refer back to previous published work. Also, more information should be provided on the potential benefit of metagenomic analyses with museum specimens. Finally, the conclusion section should be clearer with distinct recommendations for best collection, preservation and sequencing approaches with museum specimens.

c. The PCA and UMAP analyses are not included in the materials and methods section. Further, one of these ordination plots should be included in the manuscript as they show the differentiation based on sample and tissue type.

4. Higher resolution figures are required. Both Figures 1 and 3 have too low of resolution. The test is not readable in Figure 3.

Minor Comments:

1. The authors should choose a way to refer to the samples and use this consistently throughout the manuscript. The samples are sometimes referred to with the museum ID and sometimes by the year of collection.

2. On L53 the sentence “Despite this, the samples display a relatively high endogenous DNA contents, ranging from 26% to 56%, …” should be changed to “Despite this, the samples displayed a relatively high endogenous DNA content, ranging from 26% to 56%, …”.

3. On L139 the sentence “We used a silica-in-solution extraction method, which have proven highly effective …” should be changed to “We used a silica-in-solution extraction method, which has proven highly effective …”.

4. On L213 the “2000xg” should be changed to “2000×g”. This should also be changed throughout the manuscript.

5. What does the acronym LAF stand for on L217?

6. The measurement “ml” and “μl” should be changed to “mL” and “μL” throughout the manuscript.

7. On L228 the sentence should be changed from “The end-repair step was performed in 25,5 μL reactions using …” to “The end-repair step was performed in 25.5 μL reactions using …”. This change should be made throughout the manuscript.

8. On L238 the sentence “1 μL of library was qPCR quantified with Green …” should be changed to “One microliter of library was qPCR quantified with Green …” as it is the start of a sentence.

9. On L289 the authors refer to a “Supplement Table S2”. This should be changed to “Supplementary Table S2”. This change should be made throughout the manuscript.

10. Further, the names of Table S1 and S2 should be switched as the original “Table S2” is mentioned first in the manuscript.

11. The sentence starting on L365 “These snake specimens are likely to have been handled many times over the years so as proxy for DNA contamination in the various tissues, we mapped the sequencing data against the human reference genome” should also be included in the materials and methods. This is good justification as to why the human reference was used to determine the level of contamination.

12. On L382 the sentence “We focus on the overall differences in exploring …” should be changed to past tense.

13. What do the different colours represent in figure 3?

14. The section in the results starting on L401 looking at the ordination plots needs to be clearer. I agree that in the PCA especially there is differentiation based on tissue type but I am not sure it can be attributed to diversity from these analyses alone.

15. On L436 the sentence “Skin, including snake scales, are partly made of dead keratinized cells [76], why it is not unexpected that they contain lower amounts of DNA than liver tissue” is a fragment.

16. The sentence starting on L457 “Although this is still markedly lower compared to working with fresh tissue, where the endogenous DNA fraction is expected to be close to 100%, we argue that all tissues from the included specimens have adequate levels of endogenous DNA for genome-wide sequencing to be feasible” needs data or references to back up this statement.

17. On L475 “Rat” should not be capitalized.

18. In-text citations should be more consistent. For example, on L476 the sentence reads “Likewise, Allentoft, Rasmussen (13) used an aDNA …” and should be changed to reflect the requirements of PLOS ONE. This should be changed throughout the manuscript.

19. The sentence on L514 “By here mapping the sequences to a reference genome, we can measure the actual endogenous DNA content, and thus document the genomic research potential in this old formalin preserved snake from 1923” is a fragment.

20. The sentence starting on L522 “It is well known that formalin fixation may hamper or completely prevent the successful recovery of DNA from a given sample, but out results show that if indeed the DNA extraction is successful” should be changed to “It is well known that formalin fixation may hamper or completely prevent the successful recovery of DNA from a given sample, but our results show that if indeed the DNA extraction is successful”.

21. The sentence starting on L593 “Using human mitochondrial genomes as reference this was estimated to c. 1% in [80] and 2,3% in [36] and when based on genome-wide shot-gun data, values of 4.3-8.9% [94], <0,3% [35], <0,1% [34], and 0.27 % [33] have been reported” is unclear and needs to be rewritten.

22. References are needed in the paragraph starting on L603 comparing the levels of microbial DNA found in this study to other studies.

23. The sentence starting on L624 “Our metagenomic profiling is intended as a proof-of-concept to evaluate the potential and we caution that the exact taxonomic identification of specific microbial taxa require rigor follow-up investigations” should be changed to “Our metagenomic profiling is intended as a proof-of-concept to evaluate the potential and we caution that the exact taxonomic identification of specific microbial taxa require rigorous follow-up investigations”.

24. The sentence starting on L647 “This could serve as window into past microbial communities related to …” should be changed to “This could serve as a window into past microbial communities related to …”.

Reviewer #2: In their manuscript “Uncovering the genomic and metagenomic research potential in old ethanol-preserved snakes”, authors Zacho et al. describe the results of a sequencing experiment designed to evaluate 1) endogenous DNA content and preservation and 2) metagenomic data in five individuals of common garter snake (Thamnophis sirtalis) preserved in ethanol. The authors performed silica-in-solution DNA extractions and shotgun sequencing, and aligned reads to a reference genome to evaluate DNA concentration, fragment length, contamination level, and metagenomic sequence contents. They further test for the presence of formalin, a common additive to many ethanol-preserved specimens. They find extractions from bone material generally had the greatest sequence length, the highest percentage of endogenous sequence, and the lowest proportion of human contamination. However, these improvements in sequencing performance were qualitatively small. A single sample tested positive for formalin exposure, but sequencing performance was not obviously affected. Metagenomic data revealed biologically plausible, overlapping, but non-identical microbial communities among tissue types; the presence of many of these organisms in a blank control sample suggests a significant proportion represent exogenous contamination. The authors discuss the implications of these findings for museum genomics.

The paper’s main claim is that shotgun DNA sequencing can effectively retrieve endogenous genomic material from liver, bone, and skin tissue of ethanol-preserved snake specimens. The data in the paper back up this claim, though how useful these data might be in population genomic studies will depend on scope and proper bioinformatic processing. The paper also makes (hedged) claims about the performance of different tissue types, and the different microbial genetic profiles of these tissues. While framed in an appropriately descriptive way, these claims are based on small sample sizes and lack statistical justification. Writing quality is acceptable, though there are a number of minor punctuation and / or grammatical errors throughout, particularly in the introduction; I highlight some of these in my minor comments below. Figures and tables are generally clear and appropriately labeled, though the low resolution of Figure 3 in the PDF of my manuscript prevents me from evaluating its contents thoroughly.

My major suggestions are 1) adding / strengthening caveats around sample size and study conclusions and 2) carefully reading for errors in the text. Shortening the lengthy discussion might also increase the impact of the paper.

Minor comments:

L48: “Shot-gun” should be “shotgun”

L48-49: Suggested rephrase (SR): “...to test DNA sequence preservation…”

L56-58: SR: “Though at least one of the snakes had been exposed to formalin, neither the concentration nor the quality of the obtained DNA was affected.”

L61: “...invaluable source of information in the era of genomics.”

L73: morphology is also relevant (and usually sufficient) for species identification; not clear what this sentence is suggesting

L74-75: would suggest replacing the commas sandwiching the “for example” clause with parentheses

L84-85: SR: “reduce fieldwork costs and effort” or similar

L116: I suggest adding a period after the citations and beginning the next sentence with “Switching to Next Generation Sequencing”

L119-120: SR: “degraded models become a limiting factor”

L121: SR: “...it may be fruitful to apply methodological advances…”

L125: remove comma after “coverage”

L127: SR: “...represent only a minor fraction of the total quantity of sequences, the majority of which are microbial DNA.” or similar

L173: wouldn’t capitalize “Southern” in “Southern Canada”

L220: were Qubit measurements replicated for each sample? If not, I would acknowledge this—lots of variance in readings with these things

L228: replace comma in “25,5” with period

L235: SR: “...before being eluted…”

L244-245: “...ending with one minute at…”

L259: what specifically was MapDamage used for? Clarify

L271: If possible, package numbers would be great here

L388-391: long, confusing run-on sentence—consider breaking up

L490: remove comma; make sure “post-mortem” is consistently italicized throughout (c.f. L577)

L491: SR: “...arguably explain”

L492: should be “obtained the longest”

Figure 3: labels in tissue specific figures are illegible; possibly a PDF resolution issue? If not, consider replacing with a key and enlarging

6. PLOS authors have the option to publish the peer review history of their article (what does this mean?). If published, this will include your full peer review and any attached files.

Reviewer #1: No

Reviewer #2: No

---

## [Author Response · Author response to Decision Letter 0]

19 Jul 2021

Comments to the Author

Reviewer #1: Zacho et al. PLOS ONE

Uncovering the genomic and metagenomic research potential in old ethanol-preserved snakes

This manuscript examines DNA degradation in old museum preserved samples. They sample three different tissue types, skin, bone and liver, from five common garter snake (Thamnophis sirtalis) samples. These samples have been collected in different regions and different times, which allows them to examine different preservation techniques and the impacts of sample age on degradation. The authors use ancient DNA protocols and shotgun sequencing to determine the best collection, extraction and sequencing procedures for the analysis of museum specimens. Multiple different metrics, including DNA concentration, percent endogenous DNA, fragment length, C-to-T deamination damage and percent human contamination, are used and the authors find similar results across tissue types. In general, I found this manuscript to be of interest and a valuable contribution to the field. However, I did have several concerns with the manuscript that should be addressed.

Major Comments:

1. The main question of the manuscript was to determine the best approach for sequencing museum specimens and compared multiple different tissues. I believe that these analyses should be more rigorous and should include statistical analyses. The authors compare multiple metrics (Figs. 1 and 2) across tissues and specimens and the addition of statistical comparisons through an ANOVA or a similar analysis would greatly improve the manuscript. This would allow the authors to statistically compare the different tissues within a sample. Further, I believe that statistically testing for differences across samples (ie. comparing liver samples across all specimens) would also strengthen the paper. This would allow the authors to compare the ages of the samples as well as the different storage and preparation techniques used by museums.

- After careful consideration as to how we could do statistical test on our data, with different units and dependent variables, we decided to add an additional co-author (Anders Galatius) to the manuscript, since he is an expert in how to tackle such tests. The revision is therefore now updated with a linear mixed effects model, that compares the different tissue’s performance in relation to the five quality metrics. Moreover, we also include a model that compares the performance in relation to age since collection of specimens.

2. Further to the comment above, the authors should include standard deviations in the manuscript when they are using averages.

- We acknowledge that standard deviations should normally be included when using averages, but since vi are using an average from only two replicates per sample/tissue, we find it more accurate to infer the range covered by the replicates (for example as seen in Figure 1).

3. The manuscript is relatively well written, but I believe there are 3 areas that would help improve the clarity of the study:

a. The introduction is quite clear but the last 3 paragraphs need to be more concise. These paragraphs could be combined into 2 paragraphs that would be clearer. The first paragraph should address the similarities to aDNA and the availability of the metagenomic profile. The second paragraph should clearly introduce the objectives and methods of the study.

b. The discussion is quite long and needs to be shortened for clarity. Each of the sections within the discussion should be clearer and also clearly refer back to previous published work. Also, more information should be provided on the potential benefit of metagenomic analyses with museum specimens. Finally, the conclusion section should be clearer with distinct recommendations for best collection, preservation and sequencing approaches with museum specimens.

c. The PCA and UMAP analyses are not included in the materials and methods section. Further, one of these ordination plots should be included in the manuscript as they show the differentiation based on sample and tissue type.

-The last 3 paragraphs in the introduction have been reduced to 2, according to the reviewer’s suggestion. Hopefully this will make the text and aim more understandable to the reader. 

-The conclusion has been modified to be more specific regarding sampling and sequencing, whenever possible.

-We have added more information on the possible metagenomic benefits that can be possible from similar analysis

-PCA and UMAP are now explained in the materials and methods. We do not believe that these plots should be “core” figures included in the main manuscript. Since the other included figures are more informative, we suggest keeping these in the supplement. 

4. Higher resolution figures are required. Both Figures 1 and 3 have too low of resolution. The test is not readable in Figure 3.

Minor Comments:

1. The authors should choose a way to refer to the samples and use this consistently throughout the manuscript. The samples are sometimes referred to with the museum ID and sometimes by the year of collection.

-All instances are now modified, so that when mentioning year of collection, the sample ID is also provided

2. On L53 the sentence “Despite this, the samples display a relatively high endogenous DNA contents, ranging from 26% to 56%, …” should be changed to “Despite this, the samples displayed a relatively high endogenous DNA content, ranging from 26% to 56%, …”.

-Corrected

3. On L139 the sentence “We used a silica-in-solution extraction method, which have proven highly effective …” should be changed to “We used a silica-in-solution extraction method, which has proven highly effective …”.

-Corrected

4. On L213 the “2000xg” should be changed to “2000×g”. This should also be changed throughout the manuscript.

-Corrected

5. What does the acronym LAF stand for on L217?

-The acronym is replaced with the full equipment type

6. The measurement “ml” and “μl” should be changed to “mL” and “μL” throughout the manuscript.

-all instances have been changed

7. On L228 the sentence should be changed from “The end-repair step was performed in 25,5 μL reactions using …” to “The end-repair step was performed in 25.5 μL reactions using …”. This change should be made throughout the manuscript.

-Corrected

8. On L238 the sentence “1 μL of library was qPCR quantified with Green …” should be changed to “One microliter of library was qPCR quantified with Green …” as it is the start of a sentence.

-Corrected

9. On L289 the authors refer to a “Supplement Table S2”. This should be changed to “Supplementary Table S2”. This change should be made throughout the manuscript.

-Corrected for entire manuscript

10. Further, the names of Table S1 and S2 should be switched as the original “Table S2” is mentioned first in the manuscript.

-Checked that S1 is mentioned before S2

11. The sentence starting on L365 “These snake specimens are likely to have been handled many times over the years so as proxy for DNA contamination in the various tissues, we mapped the sequencing data against the human reference genome” should also be included in the materials and methods. This is good justification as to why the human reference was used to determine the level of contamination.

-Sentence with small changes are now inserted in the material and methods section, under bioinformatics.

12. On L382 the sentence “We focus on the overall differences in exploring …” should be changed to past tense.

-Corrected

13. What do the different colours represent in figure 3?

-Fig. 3 legend is updated with information on color designation

14. The section in the results starting on L401 looking at the ordination plots needs to be clearer. I agree that in the PCA especially there is differentiation based on tissue type but I am not sure it can be attributed to diversity from these analyses alone.

-The results section has been modified

15. On L436 the sentence “Skin, including snake scales, are partly made of dead keratinized cells [76], why it is not unexpected that they contain lower amounts of DNA than liver tissue” is a fragment.

-The sentence has been updated

16. The sentence starting on L457 “Although this is still markedly lower compared to working with fresh tissue, where the endogenous DNA fraction is expected to be close to 100%, we argue that all tissues from the included specimens have adequate levels of endogenous DNA for genome-wide sequencing to be feasible” needs data or references to back up this statement.

-Text is now updated with references that exemplifies why our obtained values of endogenous content are sufficient for genomics.

Verify 

17. On L475 “Rat” should not be capitalized.

-Corrected

18. In-text citations should be more consistent. For example, on L476 the sentence reads “Likewise, Allentoft, Rasmussen (13) used an aDNA …” and should be changed to reflect the requirements of PLOS ONE. This should be changed throughout the manuscript.

-references are corrected throughout the manuscript, to the format: “Allentoft et al. [13]”

19. The sentence on L514 “By here mapping the sequences to a reference genome, we can measure the actual endogenous DNA content, and thus document the genomic research potential in this old formalin preserved snake from 1923” is a fragment.

-The sentence is now modified

20. The sentence starting on L522 “It is well known that formalin fixation may hamper or completely prevent the successful recovery of DNA from a given sample, but out results show that if indeed the DNA extraction is successful” should be changed to “It is well known that formalin fixation may hamper or completely prevent the successful recovery of DNA from a given sample, but our results show that if indeed the DNA extraction is successful”.

-Corrected

21. The sentence starting on L593 “Using human mitochondrial genomes as reference this was estimated to c. 1% in [80] and 2,3% in [36] and when based on genome-wide shot-gun data, values of 4.3-8.9% [94], <0,3% [35], <0,1% [34], and 0.27 % [33] have been reported” is unclear and needs to be rewritten.

-Sentence has been rewritten

22. References are needed in the paragraph starting on L603 comparing the levels of microbial DNA found in this study to other studies.

-this was indeed needed – we have added several references in the paragraph with comparisons

23. The sentence starting on L624 “Our metagenomic profiling is intended as a proof-of-concept to evaluate the potential and we caution that the exact taxonomic identification of specific microbial taxa require rigor follow-up investigations” should be changed to “Our metagenomic profiling is intended as a proof-of-concept to evaluate the potential and we caution that the exact taxonomic identification of specific microbial taxa require rigorous follow-up investigations”.

-Corrected 

24. The sentence starting on L647 “This could serve as window into past microbial communities related to …” should be changed to “This could serve as a window into past microbial communities related to …”.

-Corrected

Reviewer #2: In their manuscript “Uncovering the genomic and metagenomic research potential in old ethanol-preserved snakes”, authors Zacho et al. describe the results of a sequencing experiment designed to evaluate 1) endogenous DNA content and preservation and 2) metagenomic data in five individuals of common garter snake (Thamnophis sirtalis) preserved in ethanol. The authors performed silica-in-solution DNA extractions and shotgun sequencing, and aligned reads to a reference genome to evaluate DNA concentration, fragment length, contamination level, and metagenomic sequence contents. They further test for the presence of formalin, a common additive to many ethanol-preserved specimens. They find extractions from bone material generally had the greatest sequence length, the highest percentage of endogenous sequence, and the lowest proportion of human contamination. However, these improvements in sequencing performance were qualitatively small. A single sample tested positive for formalin exposure, but sequencing performance was not obviously affected. Metagenomic data revealed biologically plausible, overlapping, but non-identical microbial communities among tissue types; the presence of many of these organisms in a blank control sample suggests a significant proportion represent exogenous contamination. The authors discuss the implications of these findings for museum genomics.

The paper’s main claim is that shotgun DNA sequencing can effectively retrieve endogenous genomic material from liver, bone, and skin tissue of ethanol-preserved snake specimens. The data in the paper back up this claim, though how useful these data might be in population genomic studies will depend on scope and proper bioinformatic processing. The paper also makes (hedged) claims about the performance of different tissue types, and the different microbial genetic profiles of these tissues. While framed in an appropriately descriptive way, these claims are based on small sample sizes and lack statistical justification. Writing quality is acceptable, though there are a number of minor punctuation and / or grammatical errors throughout, particularly in the introduction; I highlight some of these in my minor comments below. Figures and tables are generally clear and appropriately labeled, though the low resolution of Figure 3 in the PDF of my manuscript prevents me from evaluating its contents thoroughly.

My major suggestions are 1) adding / strengthening caveats around sample size and study conclusions and 2) carefully reading for errors in the text. Shortening the lengthy discussion might also increase the impact of the paper.

Minor comments:

L48: “Shot-gun” should be “shotgun”

-changed throughout the manuscript

L48-49: Suggested rephrase (SR): “...to test DNA sequence preservation…”

- The suggestion has been considered carefully; however we have chosen to keep the original phrase 

L56-58: SR: “Though at least one of the snakes had been exposed to formalin, neither the concentration nor the quality of the obtained DNA was affected.”

-suggestion has been incorporated in the manuscript

L61: “...invaluable source of information in the era of genomics.”

-suggestion has been incorporated in the manuscript

L73: morphology is also relevant (and usually sufficient) for species identification; not clear what this sentence is suggesting

-Sentence has been updated

L74-75: would suggest replacing the commas sandwiching the “for example” clause with parentheses

-Good point, the suggestion is integrated

L84-85: SR: “reduce fieldwork costs and effort” or similar

-suggestion is integrated

L116: I suggest adding a period after the citations and beginning the next sentence with “Switching to Next Generation Sequencing”

-the sentence has been split into two as suggested

L119-120: SR: “degraded models become a limiting factor”

-suggestion has been taken into consideration and sentence updated

L121: SR: “...it may be fruitful to apply methodological advances…”

-suggestion is integrated

L125: remove comma after “coverage”

-corrected

L127: SR: “...represent only a minor fraction of the total quantity of sequences, the majority of which are microbial DNA.” or similar

- Sentence has been updated

L173: wouldn’t capitalize “Southern” in “Southern Canada”

-Corrected

L220: were Qubit measurements replicated for each sample? If not, I would acknowledge this—lots of variance in readings with these things

-I agree, Qubit indeed produces variable results. We did measure replicates and this is now added to the text

L228: replace comma in “25,5” with period

-Corrected

L235: SR: “...before being eluted…”

-Corrected

L244-245: “...ending with one minute at…”

-Sentence 

L259: what specifically was MapDamage used for? Clarify

-Sentence has been updated with more information

L271: If possible, package numbers would be great here

-Text is updated with package numbers

L388-391: long, confusing run-on sentence—consider breaking up

L490: remove comma; make sure “post-mortem” is consistently italicized throughout (c.f. L577)

-Both points have been incorporated

L491: SR: “...arguably explain”

-Corrected

L492: should be “obtained the longest”

-Corrected

Figure 3: labels in tissue specific figures are illegible; possibly a PDF resolution issue? If not, consider replacing with a key and enlarging

-Something with the resolution was changed when converting to TIFF, which we didn’t realize at that point. The figure (before converting) has a high resolution, so we have therefore made a new version of figure 3 that should be all illegible.

---

## [Editor Report · Decision Letter 1]

5 Aug 2021

Uncovering the genomic and metagenomic research potential in old ethanol-preserved snakes

PONE-D-21-05242R1

Dear Dr. Zacho,

We’re pleased to inform you that your manuscript has been judged scientifically suitable for publication and will be formally accepted for publication once it meets all outstanding technical requirements.

Kind regards,

Christopher M. Somers

Academic Editor

PLOS ONE
---

## [Editor Report · Acceptance letter]

13 Aug 2021

PONE-D-21-05242R1 

Uncovering the genomic and metagenomic research potential in old ethanol-preserved snakes 

Dear Dr. Zacho:

I'm pleased to inform you that your manuscript has been deemed suitable for publication in PLOS ONE. Congratulations! Your manuscript is now with our production department. 

Kind regards, 

on behalf of

Dr. Christopher M. Somers 

Academic Editor

PLOS ONE